# Associations between RNA-Binding Motif Protein 3, Fibroblast Growth Factor 21, and Clinical Outcome in Patients with Stroke

**DOI:** 10.3390/jcm11040949

**Published:** 2022-02-11

**Authors:** Paulo Ávila-Gómez, María Pérez-Mato, Pablo Hervella, Antonio Dopico-López, Andrés da Silva-Candal, Ana Bugallo-Casal, Sonia López-Amoedo, María Candamo-Lourido, Tomás Sobrino, Ramón Iglesias-Rey, José Castillo, Francisco Campos

**Affiliations:** 1Clinical Neurosciences Research Laboratory (LINC), Health Research Institute of Santiago de Compostela (IDIS), 15706 Santiago de Compostela, Spain; pauloavilagomez@gmail.com (P.Á.-G.); pablo.hervella.lorenzo@sergas.es (P.H.); adopicolopez@gmail.com (A.D.-L.); ana.isabel.bugallo.casal@sergas.es (A.B.-C.); sonia.lopez.amoedo@sergas.es (S.L.-A.); maria.candamo@gmail.com (M.C.-L.); tomas.sobrino.moreiras@sergas.es (T.S.); ramon.iglesias.rey@sergas.es (R.I.-R.); jose.castillo.sanchez@sergas.es (J.C.); 2Neurological Sciences and Cerebrovascular Research Laboratory, Department of Neurology and Stroke Center, La Paz University Hospital, Neuroscience Area of IdiPAZ Health Research Institute, Universidad Autónoma de Madrid, Paseo de la Castellana 261, 28046 Madrid, Spain; mery19832005@yahoo.es; 3Neurovascular Diseases Laboratory, Neurology Service, Biomedical Research Institute (INIBIC), University Hospital Complex of A Coruña, 15006 A Coruña, Spain; andres.alexander.da.silva.candal@sergas.es

**Keywords:** body weight, clinical outcome, FGF21, RBM3, stroke, temperature

## Abstract

Background: RNA-binding motif protein 3 (RBM3) is a cold-induced marker of good functional outcome of ischemic stroke that is promising as a protective target. Fibroblast growth factor 21 (FGF21) is an obesity- and temperature-related hormone that upregulates the expression of RBM3, which is beneficial as a recombinant treatment and has been tested under different experimental pathological conditions, including stroke. However, the interaction between RBM3 and FGF21 has not yet been tested for clinical stroke conditions. Methods: In a sample of 66 stroke patients, we analyzed the associations between the FGF21 and RBM3 serum concentrations on admission and at 72 h, body weight, maximum temperature during the first 24 h, and the outcome of patients at 3 months. We also analyzed their association with biomarkers of obesity (adiponectin and leptin) and inflammation (interleukin-6 (IL-6) and interleukin (IL-10)). Results: Higher concentrations of FGF21 on admission and RBM3 at 72 h were associated with good outcomes. Serum FGF21 and RBM3 were directly related to body mass index and inversely related to the maximum temperature during the first 24 h. We found a positive association between the FGF21 concentrations in obese patients with leptin and a negative correlation with adiponectin in non-obese participants. Conclusions: This clinical study demonstrates the association between RBM3 and FGF21 levels and the outcome of stroke patients. Although further investigations are required, these data support the pharmacological induction of RBM3 as a promising protective therapy.

## 1. Introduction

Stroke is a major global health concern with devastating consequences. Its incidence and mortality rate have decreased over the past three decades; however, it is still the second leading cause of death and the leading cause of disability globally, and affects over 13 million people every year [1]. Therapeutic hypothermia is promising against the deleterious effects of acute neurological injuries, including ischemic stroke [2,3]. Hypothermia has been extensively described to be protective against ischemic injury by halting excitotoxic mechanisms, reducing infarct volume, and attenuating endothelial damage [3,4,5]. Nonetheless, the application of therapeutic hypothermia as a therapeutic tool in stroke patients remains elusive due to the associated adverse events such as shivering, coagulopathies and increased risk of infection [6,7].

Although global protein synthesis is inhibited under hypothermia, the expression of a subgroup of mediators known as cold-shock proteins (CSPs) is acutely upregulated following cold exposure [8]. In line with previous in vivo experimental studies [9,10,11,12], we have recently reported that the RNA-binding motif protein 3 (RBM3), a member of CSPs, mediates the good prognosis of ischemic stroke patients with mild body temperatures [13]. Pharmacological induction of RBM3 represents a potential means of neuroprotection for stroke in the absence of hypothermia [14]; however, a drug or agonist that directly targets RBM3 expression or activity has not been developed [14].

Fibroblast growth factor 21 (FGF21) is an integral hormone with critical regulatory functions for glycemia, lipid profile, body weight, and normothermia, among others [15]. The recombinant form (rFGF21) of FGF21 has been proposed as a novel inductor of RBM3, via downstream signaling mechanisms in cells that express the transmembrane receptor β-klotho, which is restricted to a few sites in the body, such as liver, adipose tissue, and brain [9,14]. FGF21, just like RBM3, is sensitive to cold stress and readily crosses the blood-brain barrier, and participates in the protective effect of hypothermia treatment [16,17,18,19]. In addition, FGF21 causes a slight reduction in body temperature in mammals, which is enough to induce the RBM3 response [14,20]. From a pharmaceutical point of view, it is unclear whether FGF21 mediates directly in RBM3 expression, or if it is a temperature-dependent effect. Under normothermic conditions, this hormone has been shown to be associated with body weight, as previous studies have found that obese patients have high concentrations of FGF21 [21,22]; however, changes in the concentration of RBM3 (particularly increases) under these conditions have not been analyzed as of today.

To understand the relationship between FGF21 and RBM3 in a clinical scenario such as stroke, we evaluated the associations between the circulating concentrations of both proteins and the outcome of stroke patients, and how body weight and temperature influence this response.

## 2. Materials and Methods

### 2.1. Study Design

This is a retrospective observational study using a registry of patients with ischemic stroke who were admitted to the Stroke Unit of the University Clinical Hospital of Santiago de Compostela (Spain) and were added consecutively and prospectively to our maintained databank. The study was carried out according to the principles of the Declaration of Helsinki by the World Medical Association, and approved by the Research Ethics Committee of Santiago (Project identification code 2019/616). Informed consent was obtained from each patient or from their relatives after full explanation of the procedures. The inclusion criteria were (1) availability of blood samples on admission, 72 h, and 7 days after admission (2) registered temperature measurements at 24 h, (3) registered body mass index (BMI), and (4) modified Rankin Scale (mRS) measurement at 3 months. Patients who met one of the following criteria were not considered for the study: (1) chronic inflammatory disease; (2) previous disability (defined as a score ≥2 on the modified Rankin Scale [mRS]); (3) cancer; (4) severe systemic condition that determines a life expectancy lower than 6 months; (5) infectious disease within the last 15 days; and (6) continuous anti-inflammatory drug intake within the last 15 days.

### 2.2. Clinical Variables

All patients were admitted to the stroke unit and treated using the Spanish Neurological Society protocol [21] by trained neurologists experienced in cerebrovascular diseases. Patients submitted to reperfusion therapies were treated with recombinant tissue plasminogen activator (rtPA) alone or in combination with thrombectomy. Subjects with alimentation difficulties were evaluated and followed by the Nutrition Unit from the Endocrinology Department, following the protocol in our stroke unit. The axillary temperature was recorded every 6 h by the nursing staff. Based on our previous study, we selected the maximum temperature at 24 h as an independent factor associated with the outcome of patients at 3 months [13]. For this analysis, the axillary temperature was measured at the time of admission and at 6 and 24 h. Axillary temperatures ≥37.5 °C were treated with paracetamol (500 mg p.o.) or metamizole (2 g i.v.) every 6 h. BMI was calculated using the following formula: weight (kg)/height^2^ (m^2^). It was classified according to the World Health Organization cut-off points: normal weight (BMI < 25 kg/m^2^), overweight (BMI 25–30 kg/m^2^), grade I obesity (BMI 30–35 kg/m^2^), obesity grade II (BMI 35–40 kg/m^2^ and grade III obesity (BMI > 40 kg/m^2^) [22]. The stroke subtypes were registered using the TOAST (Trial of Org 10172 in Acute Stroke Treatment) criteria [23]. The intensity of the neurological deficit was determined by the National Institute of Health Stroke Scale (NIHSS) on admission to the stroke unit, and the mRS was used to evaluate the functional deficit. Both scales were measured on admission and discharge and after 3 months of follow-up by a certified neurologist.

### 2.3. Blood Samples and Biomarker Assays

For the molecular determinations, the venous blood samples were collected diurnally in Vacutainer tubes (Becton Dickinson, San Jose, CA, USA). The blood samples were centrifuged for 10 min at 3000× *g*, and the serum was immediately aliquoted, frozen, and stored at −80 °C until analysis. The serum concentrations of RBM3 and FGF21, obesity (leptin and adiponectin) and inflammatory (interleukin-6 (IL-6) and interleukin-10 (IL-10) markers were measured using enzyme-linked immunosorbent assay (ELISA) following the manufacturer’s instructions. For the RBM3 kit (Biotez RBM3 ELISA, Berlin, Germany), the minimum assay sensitivity was 10 pg/mL with inter- and intra-assay coefficients of variation (CV) of 2.6% and 1.9%, respectively. The FGF21 assay kit (Biovendor, Brno, Czech Republic) had a minimum assay sensitivity of 7 pg/mL, with an intra-assay CV of 2% and an inter-assay CV of 3.3%. Leptin (Abnova Corporation, Taipei, Taiwan) and adiponectin (Proteintech Group, Manchester, UK) assay kits had inter- and intra-assay CVs of <8%. The IL-6 and IL-10 concentrations were measured using the IMMULITE 1000 immunodiagnostic system (Siemens Healthcare Global, Los Angeles, CA, USA).

### 2.4. Endpoints

The main endpoint of the present study was the functional outcome at 3 months represented by the mRS score, which was dichotomized into good (mRS ≤ 2) and poor (mRS > 2) outcomes. Temperature was categorized into mild hypothermia (<36.5 °C), normothermia (36.5–37.5 °C), and hyperthermia (>37.5 °C). Bodyweight was categorized as follows: normal weight, BMI < 25 kg/m^2^; overweight, BMI 25–30 kg/m^2^; grade I obesity, BMI 30–35 kg/m^2^; obesity grade II, BMI 35–40 kg/m^2^; and grade III obesity, BMI > 40 kg/m^2^. As secondary endpoints, we studied the associations between FGF21 and RBM3 and obesity (leptin and adiponectin) and inflammatory (IL-6 and IL-10) biomarkers, and their relationship with body weight and temperature.

### 2.5. Statistical Analysis

The categorical data are expressed as frequency and percentage, and the continuous data are expressed as mean (standard deviation [SD]) or median and interquartile range (25th and 75th percentiles), depending on their adjustment to normality. The normal distributions of the sample and continuous data were determined using the Kolmogorov-Smirnov test followed by a Lilliefors correction. Statistical inference was carried out with the chi-squared test, Student’s t-test, or Mann-Whitney U test, according to the nature of the contrast variable and its adjustment to normality. Bivariate correlations were determined using Pearson’s or Spearman’s correlation coefficients, depending on the distribution of the variable.

The associations of RBM3 and FGF21 on admission and at 72 h, maximum temperature at 24 h, and BMI with the outcome at 3 months were evaluated using logistic regression analysis models. Each model was adjusted for independent variables in bivariate analysis. The results were expressed as adjusted odds ratios (ORs) with respective confidence intervals of 95%. Statistical significance was set at *p* < 0.05. All analyses were conducted using SPSS Statistics (version 20.0; IBM, Chicago, IL, USA) for Mac by a researcher blinded to sample identity.

## 3. Results

### 3.1. Sample Description

Sixty-six patients from our biobank who met the inclusion criteria were included in this study. A description of the sample is provided in Table 1. Bodyweight analysis showed that 31.8% of the participants were grade I obese patients, closely followed by overweight patients (30.3%). Patients with normal bodyweight accounted for 19.7% of the sample, while the remaining 18.2% had grade II obesity. None of the patients had grade III obesity. The mean maximum axillary temperature during the first 24 h was 36.9 ± 0.6 °C.

Higher concentrations of FGF21 were found on admission, while the peak for RBM3 expression was observed 72 h after stroke onset (Figure 1a,b). Based on this analysis, we performed a correlation analysis to determine the association between the serum concentrations of FGF21 and RBM3 in both time-points (Table 2). The highest relationship was observed between the concentration of FGF21 on admission and the concentration of RBM3 at 72 h (r = 0.799; *p* < 0.0001) (Figure 2). Based on these results, the concentration of FGF21 on admission and the concentration of RBM3 at 72 h were used for the next analysis.

### 3.2. Analysis of the Association between RBM3 and FGF21 with Temperature and Body Weight

Patients with lower body temperature at 24 h showed higher concentrations of FGF21 on admission (625.1 ± 178.8 pg/mL) than normothermic patients (516.5 ± 207.5 pg/mL), while participants who developed hyperthermia showed the lowest concentrations on admission (359.9 ± 191.4 pg/mL) (Figure 3a). A similar profile was found for RBM3 at 72 h for the three temperature ranges (Figure 3b). A negative correlation was found between the maximum temperature at 24 h and the FGF21 concentration on admission (r = −0.735; *p* < 0.0001) and RBM3 concentration at 72 h (r = −0.474; *p* < 0.0001).

Regarding bodyweight, the serum concentrations of FGF21 on admission and RBM3 at 72 h were directly proportional to BMI. Grade II obesity patients had the highest circulating concentrations of FGF21 (729.6 ± 200.9 pg/mL) and RBM3 (408.8 ± 133.9 pg/mL), while patients with a BMI of <25 presented with the lowest values (FGF21 = 311.7 ± 192.9 pg/mL; RBM3 = 285.5 ± 122.2 pg/mL) (Figure 3c,d).

### 3.3. Influence of Serum RBM3 and FGF21 on Functional Outcome

Univariate analysis was performed to evaluate the differences between the groups based on the outcome (Table 3). No differences were found in the BMI or dyslipidemia. Maximum axillary temperature was significantly higher for the patients with poor outcomes (37.2 ± 0.7 °C) than for those with good outcomes (36.4 ± 0.6 °C) (*p* < 0.0001). Regarding the main biomarkers of the study, higher concentrations of FGF21 on admission were detected for the patients with good functional outcomes at 3 months than for those with poor outcomes (639.6 ± 179.9 pg/mL versus 409.7 ± 193.2 pg/mL; *p* < 0.0001) (Figure 4a). Similarly, patients with good outcomes showed higher RBM3 concentrations at 72 h as opposed to the lower concentrations of the poor outcome group (458.9 ± 95.1 pg/mL versus 252.6 ± 96.0 pg/mL; *p* < 0.001) (Figure 4b). Subsequently, we performed logistic regression to evaluate the association of clinically relevant variables with the outcome at 3 months. FGF21 on admission was independently associated with good functional outcomes at 3 months (OR 0.99, CI 95% 0.99–0.99; *p* = 0.032). A similar association was found for RBM3, as patients with higher concentrations at 72 h had better outcomes at 3 months (OR 0.97, CI 95% 0.95–0.99, *p* = 0.029). The NIHSS score on admission and axillary temperature at 24 h increased the likelihood of having a bad outcome at 3 months by 1.25 and 5 times, respectively. However, these associations were not significant when RBM3 or FGF21 was included in the model. Finally, BMI was not associated with the outcome of patients at 3 months (OR 0.92, CI 95% 0.79–1.09, *p* = 0.348).

### 3.4. Influence of RBM3 and FGF21 on Weight and Inflammation-Related Markers

The circulating concentrations of inflammation related to obesity (leptin and adiponectin) and (IL-6 and IL-10)-related biomarkers were also analyzed in this study. The concentrations of these molecules are listed in Table 4. On admission, the FGF21 concentrations were positively correlated with the IL-6 and leptin concentrations (r = 0.551, *p* < 0.0001; r = 0.677, *p* < 0.0001) and negatively correlated with the adiponectin concentrations (r = −0.536, *p* < 0.0001). Regarding RBM3, a significant correlation was found with IL-10 at 72 h (r = 0.560, *p* < 0.0001). Based on the relationship between the FGF21 concentrations and body weight, we first analyzed its association with the leptin and adiponectin concentrations.

The leptin concentrations on admission were positively correlated with FGF21 concentration and BMI (r = 0.677, *p* < 0.0001; r = 0.720, *p* < 0.0001). To enhance the statistical power of the analysis, patients were categorized by weight into non-obese (normal weight and overweight, *n* = 33) and obese (grade I and II obesity, *n* = 33). The categorized analysis of the sample showed a strong positive correlation between FGF21 and leptin concentrations on admission exclusively in obese patients (obese: r = 0.762, *p* < 0.0001 vs. non-obese: r = −0.019, *p* = 0.917) (Figure 5a).

Conversely, the adiponectin concentration was negatively correlated with FGF21 and BMI (r = −0.536, *p* < 0.0001; r = −0.763, *p* < 0.0001). Unlike leptin, when the sample was categorized by weight, the relationship between the concentration of FGF21 and adiponectin was only demonstrated for non-obese patients (r = −0.575; *p* < 0.0001) (Figure 5b).

Furthermore, we studied the association of two cytokines (IL-6) with the two biomarkers under study. On the one hand, IL-6 concentrations were significantly associated with FGF21 (r = 0.551, *p* < 0.0001) and body weight (r = 0.632, *p* < 0.0001). When the sample was arranged by weight, a similar trend was found for FGF21 and IL-6 regardless of BMI, although the association was only significant in obese patients (r = 0.367, *p* = 0.036) (Figure 5c).

In contrast, IL-10 was associated with RBM3 but not FGF21. The IL-10 concentrations at 72 h were positively associated with the RBM3 circulating concentrations at 72 h (r = 0.560, *p* < 0.0001). Mimicking RBM3 and IL-6 were inversely correlated with the maximum temperature during the first 24 h (r = −0.417, *p* < 0.0001). Following this analysis, patients were categorized based on their maximum temperature registered during the first 24 h. A similar positive correlation between IL-10 and RBM3 at 72 h was found for all patients, irrespective of the temperature range achieved within the first 24 h (Figure 5d).

## 4. Discussion

In this study, we found that higher concentrations of FGF21 on admission and RBM3 at 72 h were associated with good outcome and BMI, but inversely related to the maximum temperature during the first 24 h after stroke.

In line with our previous study [13], this new analysis reinforces the hypothesis that RBM3 acts as a promising protective target mediating the good outcome of patients with mild body temperature. These findings are not new, and the protective role of RBM3 has been widely explored and demonstrated in various scenarios, including brain injury, cancer, heart ischemia, and muscle hypertrophy, as a therapeutic target [8,9,10,11,12]. In the absence of a specific agonist, the upstream induction of RBM3 has been evaluated for its potential for use as therapy. For instance, TrkB agonism has been recently shown to induce RBM3 without cooling because of the central role of TrkB signaling in RBM3 activation; this is expected to prevent neurodegeneration damage, whereas TrkB inhibition abrogates cooling-induced RBM3 protection [24]. These new data suggest that RBM3 mediates the protective effect of hypothermia, supporting the hypothesis that pharmacological activation of RBM3 can be used therapeutically without the need for inducing hypothermia.

A well-recognized inducer of RBM3 is FGF21 [2,14]; however, the interaction between these two molecules has not been clearly elucidated.

In the field of stroke, FGF21 is known for its protective effects [25]. rFGF21 treatment leads to a reduction in inflammation and infarct volume, preserves blood–brain barrier integrity, and improves the neurological outcome in experimental animal models of cerebral ischemia [17,18,25,26,27,28,29]. In addition, due to its hormonal nature, FGF21 plays a key role in thermoregulation during cold stress by inducing non-shivering thermogenesis through white adipose tissue browning [30]. Along with the liver and pancreas, fat tissue is one of the main sources of endogenous FGF21 [15]. Diet-induced and genetically obese mice have been reported to exhibit a higher expression of FGF21 in both white adipose tissue and the liver [31,32]. In agreement with our results, clinical studies have previously shown that BMI and FGF21 concentrations are strongly correlated, and a decrease in body weight leads to a reduction in plasmatic FGF21 [33,34]. Despite the strong associations between body weight and the risks of stroke or coronary heart attack, some studies have found that overweightness is not necessarily associated with poor prognosis or higher mortality, a phenomenon known as the “obesity paradox” [35,36] in which the protection of FGF21 could play a critical role [37]. However, in this context, the involvement of RBM3 has never been evaluated in relation to FGF21 protection.

To the best of our knowledge, this is the first study to address the relationship between FGF21 and RBM3 in stroke patients and how this association relates to patient outcomes. The data from our study show that higher circulating levels of FGF21 and RBM3 on admission and at 72 h, respectively, correlate with good stroke outcomes, which is in line with previous evidence describing FGF21 as an inductor of RBM3, and with the protective effect described for both proteins under mild temperature conditions [9,14]. Higher concentrations of FGF21 on admission after stroke have also been reported in previous studies, in which the increase in FGF21 was maintained during the first 2 weeks after brain injury [38]. Other studies have also reported that serum FGF21 concentrations in patients with acute ischemic stroke were significantly higher than those in patients in the control group [39].

To elucidate whether this interaction FGF21/RBM3 was a temperature-dependent response, the same analysis was repeated considering the body weight. We found that the circulating levels of FGF21 and RBM3 on admission and at 72 h, respectively, were strongly correlated with BMI. Moreover, the highest and lowest concentrations of both proteins were found in obese and lean patients, respectively. The above findings suggest that lower body temperatures can influence RBM3 expression directly or indirectly via FGF21 signaling, and the reported increase in RBM3 expression related to body weight may be attributed to the increased concentrations of FGF21. From a clinical perspective, it is worth noting that an increase in RBM3 may not require a reduction in body temperature, and that its expression could be induced through pharmacological approaches, such as FGF21 or TrkB agonism, as previously described [24].

Our findings reveal that FGF21 concentrations in obese patients are also correlated with other factors, such as leptin or adiponectin; therefore, it is possible that other metabolic mechanisms contribute to its protective effect. For instance, previous evidence has shown that acute administration of leptin increases circulating levels of FGF21 [40]. In pre-clinical studies, both compounds have been shown to effectively protect against cerebral ischemia by reducing brain edema, infarct volume, and macrophage infiltration, along with improving the associated neurological deficits [26,41,42].

Regarding the inflammatory markers, the chronic inflammatory state in obesity could explain the increased levels of IL-6 and their association with FGF21, as roughly 30% of circulating IL-6 is derived from adipose tissue [43]. We also believe that the association of RMB3 and IL-10 is an original contribution to the neuroprotective mechanism associated with low temperatures. IL-10 is generally known as an anti-inflammatory cytokine that exerts a plethora of immunomodulatory functions during an inflammatory response and is particularly important during the resolution phase of stroke [44]. The association observed between RMB3 and IL-10 is novel from a therapeutic perspective, although the exact pathways by which these molecules interact needs further analysis.

One important consideration in this study is that RBM3 and β-klotho receptors (required for FGF21 signaling) are abundant during development and in the neonatal brain, but low or absent in the adult brain [9,45]. RBM3 has been previously detected in the hippocampus and prefrontal cortex brain samples from humans of different ages (<1 to 35 years), and the findings suggest that the main protective efficacy of RBM3 is against neonatal brain injury. By contrast, in other studies, RMB3 was detected in brain tissue from adult rats and mice [13,26]. In our previous analysis [13] and this new study, RBM3 was detected in blood using a commercial ELISA kit, and the mean age of the stroke patients included was >50 years. Extra-cerebral effects of temperature or FGF21 in the context of obesity could also induce RBM3 release from other organs, and would be the more biologically plausible explanation of the increase in RBM3 in adult patients, thus providing new insight into the obesity paradox.

The therapeutic potential of FGF21 in stroke has been indicated by studies in animal models [25]; however, the main limitation for a future translational application is the side effects of other related factors, such as FGF2 and FGF23, in clinical trials with stroke patients [25]. Therefore, artificial RBM3 agonists may be useful alternatives to FGF21 for the treatment of ischemic vascular diseases.

The present study has some limitations. This is a retrospective study that was conducted on a small sample with a short follow-up period, which limits the number of analyses for multiple comparisons. Additional samples from different time-points during the acute phase after stroke (i.e., 24 h or 48 h) would allow for a better protein expression profile in serum. A healthy control group to determine the baselines of both proteins would significantly improve the value of the analysis and show whether stroke injury affects the expression of both proteins. Furthermore, the potential impact of the application of therapeutic hypothermia or targeted temperature management in stroke therapy on the CSPs was also not examined. This could be important, given that in pre-clinical studies, the ability of FGF21 to induce RBM3 was specifically shown in the context of mild hypothermia. Finally, our study did not include any grade III obesity patients, but the trend in our sample suggests that FGF21 and RBM3 would increase accordingly.

## 5. Conclusions

Our study shows the association of FGF21 and RBM3 on the prognosis of ischemic stroke patients, and supports the development of new pharmacological tools for RBM3-mediated neuroprotection in the absence of hypothermia.

## Figures and Tables

**Figure 1 jcm-11-00949-f001:**
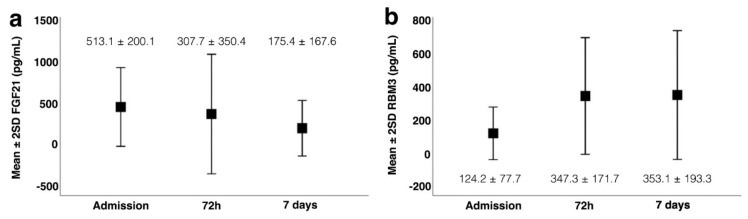
Profile level of fibroblast growth factor 21 (FGF21) (**a**) and RNA-binding motif protein 3 (RBM3) (**b**) on admission, 72 h and 7 days.

**Figure 2 jcm-11-00949-f002:**
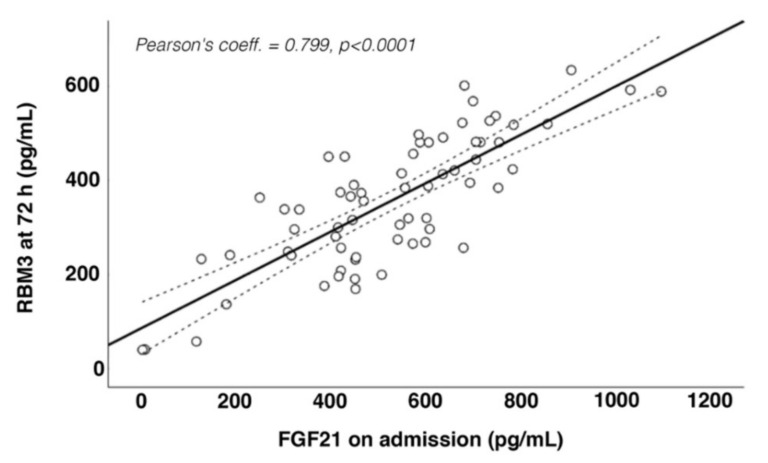
Association between fibroblast growth factor 21(FGF21) on admission and RNA-binding motif protein 3 (RBM3) at 72 h.

**Figure 3 jcm-11-00949-f003:**
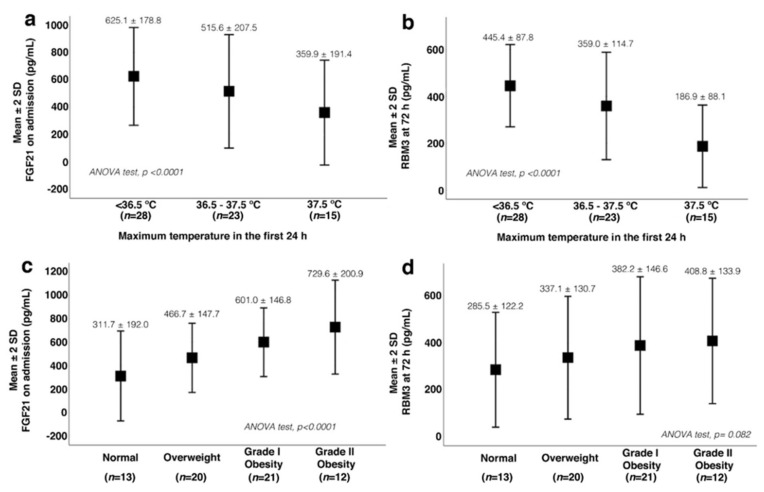
Analysis of serum fibroblast growth factor 21 (FGF21) and RNA-binding motif protein 3 (RBM3) and their association with temperature (**a**,**b**) and body mass index (BMI) (**c**,**d**).

**Figure 4 jcm-11-00949-f004:**
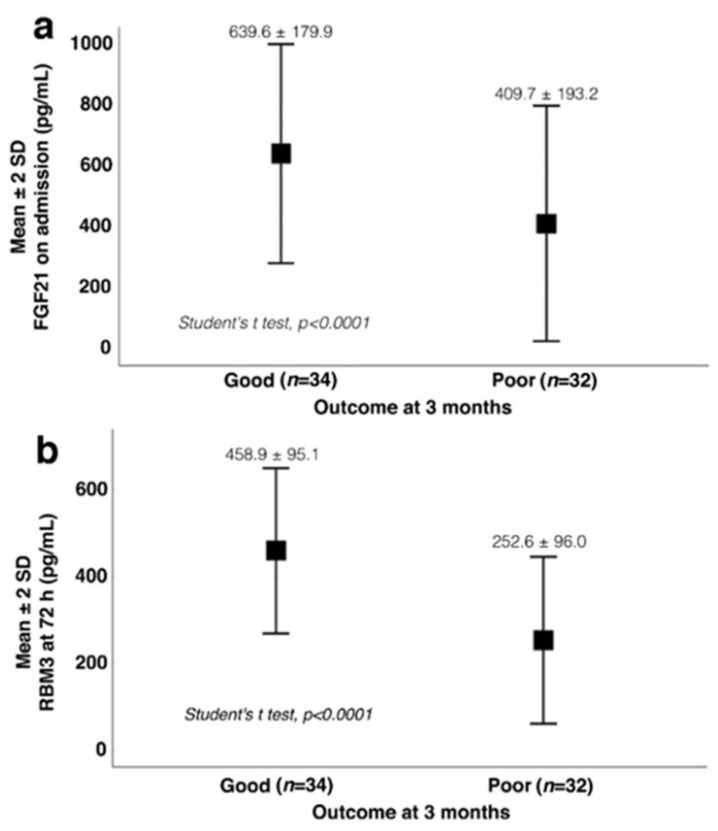
Analysis of serum fibroblast growth factor (FGF21) (**a**) and RNA-binding motif protein 3 (RBM3) (**b**) and their associations with the functional outcomes of patients.

**Figure 5 jcm-11-00949-f005:**
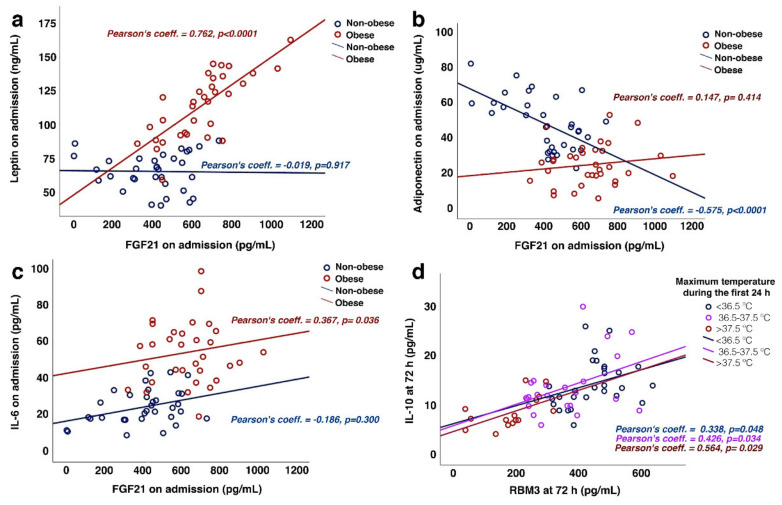
Association of fibroblast growth factor 21 (FGF21) and RNA-binding motif protein 3 (RBM3) with obesity and inflammatory biomarkers. (**a**–**c**) Association between FGF21 on admission and leptin (**a**), adiponectin (**b**), and interleukin-6 (IL-6) (**c**) concentrations in obese and non-obese participants. (**d**) Association of the circulating concentrations of RBM3 and interleukin-10 (IL-10) at 72 h with the maximum temperature during the first 24 h.

**Table 1 jcm-11-00949-t001:** Description of the 66 obese stroke patients analyzed in the present work. BMI: body mass index; TOAST: Trial of Org 10172 in Acute Stroke Treatment; NIHSS: National Institutes of Health Stroke Scale; mRS: modified Rankin Scale.

	Variable
Age (years)	67.6 ± 14.6
Female gender (%)	45.5
Arterial hypertension (%)	47.0
Diabetes (%)	28.0
Dyslipemia (%)	37.9
Smoking (%)	22.7
Alcohol consumption (%)	15.2
Atrial fibrillation (%)	33.3
Weight: -Normal (BMI < 25 kg/m^2^) (%)	19.7
-Overweight (BMI 25–30 kg/m^2^) (%)	30.3
-Obesity grade I (BMI 30–35 kg/m^2^) (%)	31.8
-Obesity grade II (BMI 35–40 kg/m^2^) (%)	18.2
Axillary temperature on admission (°C)	35.7 ± 0.6
Maximum temperature in the first 24 h (°C)	36.9 ± 0.6
Maximum temperature in the first 24 h (°C) categorized (%):	
-<36.5 °C	31.8
-36.5–37.5 °C	48.5
->37.5 °C	19.7
TOAST (%):	
-Atherothrombotic	10.6
-Cardioembolic	37.9
-Lacunar	6.1
-Indeterminate	45.4
Reperfusion treatment (%)	42.0
NIHSS on admission	9 [3,4,5,6,7,8,9,10,11,12,13,14,15]
mRS at 3 months	2 [1,2,3,4]
Outcome at 3 months (%):	
-Good	51.6
-Poor	48.4

**Table 2 jcm-11-00949-t002:** Correlation of fibroblast growth factor 21 (FGF21) and RNA-binding motif protein 3 (RBM3) on admission and at 72 h.

	pg/mL		pg/mL	Pearson’ Coefficient	*p*
RBM3 on admission	361.8 ± 161.1	FGF21 admission	526.7 ± 215.3	0.409	0.001
		FGF21 72 h	310.3 ± 282.4	−0.015	0.906
RBM3 at 72 h	356.5 ± 139.3	FGF21 admission	526.7 ± 215.3	0.799	<0.0001
		FGF21 72 h	310.3 ± 282.4	0.040	0.747

**Table 3 jcm-11-00949-t003:** Univariate analysis of the sample.

	Good Outcomen *n* = 34	Poor Outcomen *n* = 32	*p*
Age (years)	59.3 ± 13.0	76.4 ± 10.5	<0.0001
Female gender (%)	41.2	50.0	0.319
Arterial hypertension (%)	38.2	56.3	0.111
Diabetes (%)	15.6	41.2	0.021
Dyslipemia (%)	38.2	37.5	0.576
Smoking (%)	35.3	9.4	0.012
Alcohol consumption (%)	17.6	12.5	0.407
Atrial fibrillation (%)	20.6	46.9	0.022
Body mass index (kg/m^2^)	30.2 ± 5.6	29.1 ± 5.1	0.358
Weight: -Normal (BMI < 25 kg/m^2^) (%)	17.6	21.9	0.941
-Overweight (BMI 25–30 kg/m^2^) (%)	29.3	31.3	
-Obesity grade I (BMI 30–35 kg/m^2^) (%)	32.4	31.3	
-Obesity grade II (BMI 35–40 kg/m^2^) (%)	20.6	15.6	
Maximum temperature in the first 24 h (°C)	36.4 ± 0.6	37.2 ± 0.7	<0.0001
Maximum temperature in the first 24 h (°C) categorized (%):			<0.0001
-<36.5 °C	61.8	21.9	
-36.5–37.5 °C	35.3	34.4	
->37.5 °C	2.9	43.8	
TOAST (%):			0.069
-Atherothrombotic	11.8	9.4	
-Cardioembolic	20.6	56.3	
-Lacunar	8.8	3.1	
-Indeterminate	58.7	31.3	
Reperfusion treatment (%)	32.3	53.1	0.072
NIHSS on admission	3 [1,2,3,4,5,6,7,8,9]	14 [10,11,12,13,14,15,16,17,18,19]	<0.0001
RBM3 at 72 h (pg/mL)	458.9 ± 95.1	252.6 ± 96.0	<0.0001
FGF21 on admission	639.6 ± 179.9	409.7 ± 193.2	<0.0001

BMI: body mass index; TOAST: Trial of Org 10172 in Acute Stroke Treatment; NIHSS: National Institutes of Health Stroke Scale; RBM3: RNA-binding motif protein 3; FGF21: fibroblast growth factor 21.

**Table 4 jcm-11-00949-t004:** Concentrations of biomarkers on admission and at 72 h.

	Admission	72 h	*p*
IL-6 (pg/mL)	39.71 ± 20.63	43.34 ± 28.00	0.035
IL-10 (pg/mL)	11.92 ± 6.85	13.01 ± 5.38	<0.0001
Leptin (ng/mL)	89.82 ± 31.23	83.82 ± 28.77	<0.0001
Adiponectin (μg/mL)	36.81 ± 17.91	30.84 ± 15.95	<0.0001

IL-6: Interleukin 6; IL-10: Interleukin 10.

## Data Availability

All data are available in the text of the manuscript. Further anonymized data can be made available to qualified investigators upon reasonable request.

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
