# Peer review of "Associations between RNA-Binding Motif Protein 3, Fibroblast Growth Factor 21, and Clinical Outcome in Patients with Stroke"

_jcm, 2022, doi:10.3390/jcm11040949_

Round 1
Reviewer 1 Report
COMMENTS TO THE AUTHORS
This is study of the CSPs RBM3 and FGF21 in 66 adult patients with stroke managed by neurocritical care specialists in Spain. Relationships between these two CSPs and between them and body temperature, body mass index, obesity biomarkers, and serum cytokines were explored. Higher concentrations of FGF21 on admission and RBM3 at 72 h were associated with favorable outcome and BMI but inversely related to maximum temperature during the initial 24 h after stroke. These correlations provide some provocative clinical support for the recently described ability of FGF21 to induce the neuroprotective CSP RBM3 in vitro. The relationships between these CSPs and obesity parameters and markers of inflammation after stroke are also novel. Overall, despite the retrospective design, this is a novel clinical report on a cutting-edge topic in neurocritical care—the potential role of CSPs in hypothermic neuroprotection. However, the findings are greatly overstated and need to be toned down since causality has not been shown. The manuscript has other concerns and limitations which will also require a major revision along with the need for some grammatical revisions.
Critique
- The title is greatly overstated. You have not proven cause and effect. A more appropriate title would be “Associations between RNA-binding motif 3, fibroblast growth factor 21, and clinical outcomes in patients with stroke.” Please see Leisman et al, Crit Care Med, 2020 for guidance on interpretation of prediction models, where causal inference cannot be claimed. Also, the common nomenclature for RBM3 is “RNA-Binding Motif 3” and should be used instead of “ribonucleic acid-binding protein 3”.
- The results section of the abstract should state the number of subjects.
- The conclusions of the abstract are also greatly overstated. You did not demonstrate that RBM3 mediates the effect of FGF21. You also did not demonstrate that FGF21 has any “effect.” Rather, you show associations between FGF21 and RBM3, and associations between both CSPs and stroke outcome—along with associations between these CSPs and both obesity and obesity markers, and serum cytokines. You did not demonstrate an “effect” rather, only associations. Your study does not support the pharmacological induction of RBM3 in the absence of hypothermia. In Fig 2, the highest levels of FGF21 and RBM3 in the patients were seen in the presence of hypothermia! It would be more accurate to state that your findings warrant future investigations on the role of CSPs FGF21 and/or RBM3 in stroke.
- In the introduction you state that “the use of therapeutic hypothermia in patients remains elusive”. TH is standard of care in cardiac arrest, perinatal asphyxia, cardiopulmonary bypass, and other conditions. Presumably, you are referring to the lack of clinical evidence-based trials supporting the use of TH in stroke. This should be made clear, and statements that over generalize should be avoided.
- In the introduction you state that “FGF21…expression is primarily restricted to the liver and adipose”. In the context of the paragraph this sentence implies that the activity of FGF21 is restricted to those organs because FGF21 production is restricted. However, the restricted activity of FGF21 is mainly due to the limited expression of β-klotho, which I think is what you mean to say. Please revise this statement for clarity.
- In the introduction you reference Inagaki et al. which showed that FGF21 decreased temperature in mice. They used transgenic mice which resulted in life-long supra-high FGF21 levels. Those mice manifested a chronic state of ketogenesis which may have contributed to the temperature decrease. You referenced this study to support the conclusion that increased FGF21 levels in some patients with lower temperatures was driven by FGF21. However, the reverse is equally likely (or perhaps even more likely); mild hypothermia stimulates FGF21 upregulation to activate non-shivering thermogenesis and promote body warming. This possibility needs to be considered.
- No information was provided on the management of these patients. Was temperature control provided and if so, how? What was used as “reperfusion treatment” in the 42% of patients who received it? Did patients in the “37.5°C” cohort receive anti-pyretic therapy (e.g., acetaminophen) or other therapies to maintain normothermia? A brief paragraph on neurocritical care management should be provided.
- Further details about blood samples, time and testing are needed. What was median time to admission sample? Did all 66 patients have both timepoints available? Were there other samples within the first 72 h that were tested? Prior studies in healthy humans have shown diurnal variation in FGF21 levels with peaks at 0800 (Lee, JCEM, 2013). Were all samples tested from a set time of day or did it vary?
- Under statistical analysis, the author’s state that the researcher was blinded to randomization – no randomization was performed.
- Box plots of patient temperature for the three groups (<36.5°C° vs. 5°C-37.5°C ° vs. 37.5°C) need to be shown with minimum, maximum, median, and IQR. Otherwise, it is unclear what the statement “maximum <36.5°C indicates”? For example, any maximum <36.5°C (e.g., 34°C, 35°C, 36°C) would fall into this sub-group but the absolute difference in temperature may be important). Also, in the graphs in Fig 2, please state “>37.5°C”, as was described under Section 2.4 Endpoints. Presumably, some of the patients in the “>37.5°C group” may have had a fever (e.g., 38°C). Simply indicating “37.5°C” without the addition of the “greater than sign” (>) implies patient temperatures were clamped, which was not the case.
- A table should be included early in the manuscript (after Table 1) which shows the ELISA values for “admission” and “72h” for FGF21 and RBM3 (i.e., akin to Table 4 for the obesity-related biomarkers). It is unclear how FGF21 and RBM3 levels changed across the two time points. Alternatively, in your prior paper on this topic, you calculated delta RBM3 at 72h (Avila-Gomez, Brain Communications, 2020). If, you claim, initial FGF21 levels impact 72h RBM3 levels, then I would examine delta RBM3 rather than absolute levels. Also, the use of the 72h timepoint should be justified – is this the last timepoint available? Is there a reason that RBM3 should peak at 72h or later? Why not a 24 or 48-h analysis?
- The authors state higher concentrations of FGF21 were seen in hypothermic patients compared to normothermia patients, but only provide the results of the ANOVA. What were the results of the post-hoc test for individual comparisons? These should either be reported in the text or in Fig 2a. These results are also needed for Fig 2b.
- Under section 3.3, the manuscript states that “FGF21 on admission was independently associated with good functional outcomes at 3 months (OR 0.99, CI 95% 0.99-0.99; p=0.032).” Are these numbers correct?
- In several of the tables you use the term “basal” when referring to RBM3 or obesity-related biomarkers. Basal implies “baseline” levels of serum biomarkers prior to an insult, which differs from “admission” levels collected after stroke. Please revise to indicate “admission” levels. Notably, the lack of an uninjured control group to ascertain a true “baseline” comparison group should also be mentioned as a limitation.
- In many of the figs, the number of patients in each group needs to be stated in the fig legends. For instance, in Fig 2A, how many patients are included in each of the <36.5°C° vs. <36.5°C-37.5°C ° vs. 37.5°C groups?
- A table with the results of the multivariable analyses should be provided.
- Several points in the Discussion are overstated and must be revised to avoid overinterpreting the results:
- Page 10, line 261…These new data suggest…rather than These new data demonstrate…
- Page 10, line 303…The above findings suggest that lower body temperatures can influence RBM3 expression…rather than…Based on the above findings, lower body temperatures can influence…
- Page 10, line 307…an increase in RBM3 may not require…rather than…an increase in RBM3 does not require…
- Page 11, line 311…it is possible that other metabolic mechanisms…rather than…it is likely that other metabolic mechanisms…
- On page 11, line 313, it would be more accurate to state…In pre-clinical studies, both markers have been shown to effectively…rather than…In stroke, both markers have been shown to effectively…
- Given the role of important brain regions such as the prefrontal cortex and hippocampus in stroke and recognizing the findings of Jackson et al [9] in the human brain in newborns and infants but not adults, it is difficult to believe that high levels and or inducible increases in RBM3 and/or the FGF21 receptor βKlotho in other regions of the adult brain represent the reason for the findings in this study. It is possible that beyond infancy, extracerebral effects of these CSPs could play an important role in stroke in adults, and thus correlated with favorable outcomes. That concept is not discussed and is more biologically plausible than the explanations discussed. Extra-cerebral effects would also be potentially more likely in the setting of obesity, which was revealed by your work to play a potential role in this pathway and in providing new insight into the obesity paradox. The discussion on these points should be revised.
- The authors performed a detailed analysis of inflammation and FGF21/RBM3, but fail to discuss the implications of this analysis, and make only passing reference to their analysis of FGF21/RBM3 and obesity.
- An additional limitation of the paper was the number of analyses without controlling for multiple comparisons. This is to be expected with this type of paper but should be acknowledged in the manuscript.
- Page 10, line 4, you reference your prior study, but the reference is a to an animal study. The pre-clinical nature of the referenced study should be acknowledged in the text. Alternatively, the authors may have intended to reference paper 13.
- In the limitations, please state that “Also the potential impact of the application of therapeutic hypothermia or targeted temperature management in stroke therapy on the CSPs was also not examined. It could be important given that in pre-clinical studies, the ability of FGF21 to induce RBM3 was specifically shown in the setting of mild hypothermia [14].
- The conclusion should be modified. Again, you did not show cause and effect, only associations. You show associations between serum FGF 21 and RBM3, and associations between both CSPs and stroke outcome. You did not demonstrate an “effect” rather, only associations.
- You need to mention that the inability to measure brain tissue RBM3 levels is an important study limitation. The experimental design does not address if serum FGF21 levels are associated with an increase in brain tissue RBM3 levels. Do the authors have a hypothesis as to the source of RBM3 in the serum? For instance, is it from cold-induced upregulation in white blood cells or from release by damaged tissues?
- Table 3 indicates that patients with higher admission FGF21 levels were associated with a favorable outcome. The significance of this finding needs to be more thoroughly discussed. Do you think that early FGF21 supplementation may be beneficial? Otherwise, might increased FGF21 levels serve as a biomarker of fewer chronic comorbidities (i.e., advanced age, lower incidence of diabetes, and atrial fibrillation), which contributed to a favorable outcome?
Grammatical concerns
- It would be more appropriate to use favorable and unfavorable outcome throughout the manuscript, figures, and legends rather than “Good” and “Poor.”
- On page 2 line 50, presumably you meant to say …patients with mild reductions in body temperature, rather than “mild body temperature.
- On page 9, line 240, the word Similarly would be preferred to the word “Mimicking.”
- On page 10, line 261, mild hypothermia (or mildly reduced body temperature) should be used rather than mild body temperature.
- On page 10, line 261…and RBM3 has been…rather than …and the RBM3 has been…
- On page 10, line 294…under mild temperature reductions (or under mild hypothermia)…rather than…under mild temperature conditions…
- Reference 26 is cited incompletely.
- Several times throughout the manuscript (e.g., paragraph 5 in the discussion) the authors state “Inductor of RBM3”. It should be “inducer of RBM3”.
Reviewer 2 Report
The authors presented an interesting approach aiming to show the potential utility of addressing a new methabolic pathway as target for neuroptotection in obese stroke patients. I think that the conclisions could be expressed in a more cautious manner but the data are priomising.
